# Topological states in double monolayer graphene

**Ying-Hai Wu⋆**

School of Physics and Wuhan National High Magnetic Field Center,
Huazhong University of Science and Technology, Wuhan 430074, China

⋆ yinghaiwu88@hust.edu.cn

## Abstract

Motivated by the experiments on double monolayer graphene that observe a variety of fractional quantum Hall states [Liu *et al.*, Nat. Phys. 15, 893 (2019); Li *et al.*, Nat. Phys. 15, 898 (2019)], we study the special setting in which two monolayers have different areas. It has not been considered before and allows us to construct a class of exotic topological states. The elementary excitations of these states do not carry fractional charges but obey fractional statistics. This is in sharp contrast to all previously studied cases, where the two properties are intimately connected and serve as hallmarks of fractional quantum Hall states. Numerical calculations are performed to demonstrate that some states can be realized with realistic parameters.



# 1  Introduction

Quantum Hall states are prototypical examples of topological states in two dimensions [1, 2]. The longitudinal conductance is exponentially suppressed as temperature decreases due to the energy gap in the bulk and the transverse Hall conductance is precisely quantized to certain rational values (in units of $e^2/h$). For integer quantum Hall (IQH) states, a free fermion picture relying on the topological Chern invariant [3] is adequate for most purposes. The states with fractional Hall conductance are much more difficult to understand as they only arise if strong electron-electron interactions are present. Based on very general principles, it can be shown that fractional Hall conductance leads to elementary excitations with fractional charges and obey fractional statistics [4, 5]. The reverse deduction is false because fractional charge can also appear in a system with integral Hall conductance. The existence of fractionalization underlies the concept of topological order as they imply that topological structure of the system affect some global properties [6].

The synthesis of graphene and other 2D materials significantly advance the studies on quantum Hall physics [7–27]. Because of their intrinsic 2D nature, one can access electrons directly in such systems and measure certain quantities besides electric transports. An intricate pattern of FQH states have been observed which reflects the interplay of spin, valley, and layer degree of freedoms. This work is primarily motivated by Refs. 25, 26 that investigated double monolayer graphene. The two monolayers are separated by hexagonal boron nitride (hBN) so direct tunneling is forbidden but interlayer Coulomb repulsion persists. Although similar structures have been made using conventional semiconductors [28–31], many new states have been observed in graphene. This system has also been studied in other recent theoretical works [32, 33].

The single-particle states of two-dimensional electrons in a perpendicular magnetic field are discrete Landau levels (LLs). If one LL is partially occupied by free electrons, there is a huge degeneracy associated with putting electrons into the single-particle states. It is remarkable that interactions would generate gapped FQH states at certain rational filling factors (i.e., the number of electrons divided by the number of states in each LL). A large number of them can be explained using the composite fermion theory [34]. The essence of this theory is to map strongly correlated states of electrons to uncorrelated states of emergent particle-flux bound states called composite fermions. After the flux attachment process, the effective magnetic field experienced by the composite fermions is different from the actual magnetic field. Many-body states of composite fermions can be constructed easily because they can be treated as non-interacting to a very good approximation. If they form an IQH state, the corresponding state of electrons would be an FQH state. It is also possible that the composite fermions form a Fermi liquid (when the effective magnetic field for them vanishes), which results in a non-Fermi liquid state of electrons [35, 36]. FQH states in monolayer graphene have been studied using the composite fermion theory and quantitative comparisons with experiments have been performed [37, 38].

The states observed in double monolayer graphene can be accounted for by a flux attachment pattern in which one electron in a particular monolayer is dressed with two fluxes from the same monolayer and one flux from the other monolayer [25, 26]. In this paper, we consider the special setting in which the two monolayers have *different* areas. To the best of our knowledge, this has not been considered before. We propose a class of topological states for which the elementary excitations have no fractional charge but still obey fractional statistics. This is quite surprising because fractional charge and fractional statistics are generally believed to be concomitant in FQH states. In some cases, the fact that elementary excitations carry fractional charges can even be used to prove that they obey fractional statistics [39, 40].

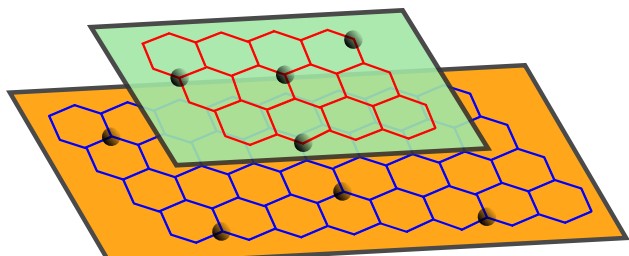

Figure 1: Schematic of the double monolayer graphene system with different areas. The number of electrons in the top (bottom) monolayer is $N_e^t$ ($N_e^b$). One electron in a specific monolayer is attached with two fluxes from the same monolayer and one flux from the other monolayer.

## 2 Models

The system of interest to us is depicted in Fig. 1. It contains two graphene monolayers in the $x$-$y$ plane that we call *top* and *bottom*. The symbols $t$ or $b$ are attached to physical quantities to indicate that they are associated with the respective monolayers. Their areas depend on the topological state that one hopes to realize and will be explained in detail later. An external magnetic field is applied in the vertical $z$ direction and we focus on the zeroth LL. In spite of the spinor nature of graphene, the wave functions in the zeroth LL have only one nonzero component that are exactly the same as those in the non-relativistic lowest LL. For the planar disk in the symmetric gauge, the explicit solutions (without normalization) are $\sim z^m \exp(-|z|^2/4)$, where $z = (x + iy)/\ell_B$ is the holomorphic coordinate and $\ell_B = \sqrt{\hbar c/(eB)}$ is the magnetic length. It is assumed the electrons are spin and valley polarized in both layers. This is the case in many previous experiments and a reasonable starting point for our investigation. One can expect to observe other interesting states if this constraint is relaxed.

Let us denote the number of electrons in the two monolayers as $N_e^t$ and $N_e^b$, which are taken to be the same in the ground states. This is a convenient but not essential choice. It is expected that the states to be discussed can also be realized in unbalanced monolayers. The kinetic energy is a constant that may be neglected. The electrons interact with each other through the Coulomb potential

$$V_{\sigma\tau}(\mathbf{r}_1 - \mathbf{r}_2) = \frac{e^2}{\varepsilon\left[|\mathbf{r}_1 - \mathbf{r}_2|^2 + (1 - \delta_{\sigma\tau})D^2\right]^{1/2}}, \tag{1}$$

where $\sigma, \tau = t, b$ refer to the monolayers, $D$ is the separation between the monolayers, and $\varepsilon$ is the dielectric constant. The energy is measured in units of $e^2/(\varepsilon\ell_B)$. The dielectric constants of graphene and hBN are different, but it is sufficient to use a single parameter $\varepsilon$ for our current purpose. If comparison with experiments is desired, one can simply rescale $D$ to account for the difference. In numerical calculations, the system is placed on the sphere [41] to mitigate finite-size edge effects. A magnetic field along the radial direction of the sphere is generated by a magnetic monopole at its center [42]. For the special setting that we shall explore, it is necessary to define separate magnetic fluxes for the top and bottom monolayer as $N_\phi^t$ and $N_\phi^b$, respectively. The number of electrons and the number of fluxes are related by $N_\phi^t = N_e^t/\nu^t - S^t$ and $N_\phi^b = N_e^b/\nu^b - S^b$, where $\nu^t, \nu^b$ are filling factors in the thermodynamic limit and $S^t, S^b$ are $O(1)$ numbers called shift. In the second quantized notation, the many-body Hamiltonian is

$$\frac{1}{2} \sum_{\sigma\tau} \sum_{\{m_i\}} F_{m_1 m_2 m_4 m_3}^{\sigma\tau\tau\sigma} C_{\sigma,m_1}^\dagger C_{\tau,m_2}^\dagger C_{\tau,m_4} C_{\sigma,m_3}, \tag{2}$$

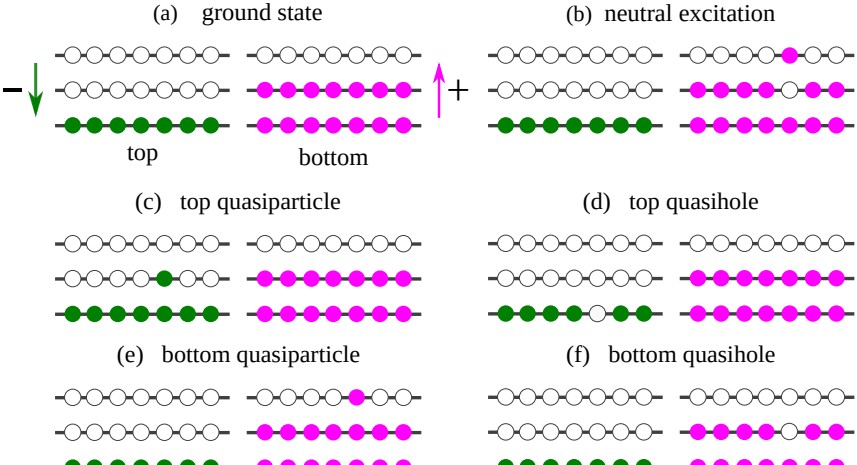

Figure 2: Schematics of the composite fermion configurations of the $\Psi^{221}_{-1,2}$ ground state and elementary excitations. The electrons in the top and bottom monolayers are converted to their respective composite fermions. One species experiences negative effective magnetic field and the other experiences positive effective magnetic field.

where $C^{\dagger}_{\sigma,m}$ ($C_{\sigma,m}$) is the creation (annihilation) operator for the single-particle state indexed by $m$ in the $\sigma$ monolayer. The coefficients $F^{\sigma\tau\tau\sigma}_{m_1 m_2 m_4 m_3}$ can be evaluated using the single-particle wave functions on the sphere and the potential $V_{\sigma\tau}(\mathbf{r}_1 - \mathbf{r}_2)$ as explained in the Appendix. Exact diagonalization (ED) and density matrix renormalization group (DMRG) are employed to compute the low-lying eigenstates of the system. If the Hilbert space dimension is not too large ($\sim 10^9$), sparse matrix diagonalization can be carried out to obtain a few low-energy eigenstates. DMRG is a vartional algorithm that searches for the ground state of a Hamiltonian within the class of matrix product states [43,44]. Its utility in studying quantum Hall physics has been demonstrated in previous works [45–53]. The long-range Coulomb interaction is handled using the method of Ref. 54.

## 3 Results

The flux attachment pattern observed in experiments [25,26] motivates the topological states to be discussed here. There are two types of composite fermions moving in two effective magnetic fields. If the two monolayers have different areas, the total number of fluxes in the two monlayers would be different. It is possible to adjust the system parameters such that the effective magnetic fields in the two monolayers have *opposite* directions. In particular, the number of fluxes in the top monolayer would be fixed at $N^t_e + N^b_e$ and that in the bottom monolayer is varied to produce a class of states. After performing flux attachment, the effective magnetic fluxes for the composite fermions in the top (bottom) monolayer is $\widetilde{N}^t_\phi$ ($\widetilde{N}^b_\phi$). For the top monolayer, the total number of fluxes attached to the electrons are $2N^t_e + N^b_e$ so we have $\widetilde{N}^t_\phi = -N^t_e$, and the composite fermions form an IQH state. If the composite fermions in the bottom monolayer also form an IQH state, the corresponding state of electrons are expected to be a gapped topological state.

To analyze the properties of these states, it is very helpful to write down explicit wave functions on the plane, which can be converted to the sphere in a straightforward manner if

needed [41]. The ground states are captured by

$$\Psi_{-1,n}^{221} \sim \left[ \Phi_{-1}^t(\{z_j^t\}) \Phi_n^b(\{z_j^b\}) \right] \prod_{j<k}(z_j^t - z_k^t)^2 \prod_{j<k}(z_j^b - z_k^b)^2 \prod_{j,k}(z_j^t - z_k^b), \tag{3}$$

where $\Phi_{-1}^t(\{z_j^t\})$ is the $\nu = 1$ IQH state of composite fermions in the top monolayer, $\Phi_n^b(\{z_j^b\})$ is the $\nu = n$ IQH state of composite fermions in the bottom monolayer, and the product factors implement the flux attachment. The case with $n = 2$ is illustrated in Fig. 2 (a). The $\sim$ sign is used because the Gaussian factor and the zeroth LL projection are omitted. The filling factors are $\nu^t = 1/2$, $\nu^b = n/(3n+1)$ and the shifts are $S^t = 1, S^b = n + 2$. It has been verified that Eq. (3) with $n = 1, 2$ provides excellent approximations to the ground states obtained by exact diagonalization in many cases [see Fig. 3 (a) and Fig. 4 (a)]. For $N_e^t = N_e^b = 6$ and $D = 0.7$, the overlap between the trial wave function and the exact eigenstate is 0.9958 (0.9897) at $n = 1$ ($n = 2$). As one would expect from the flux attachment picture, the overlap first increases and then decreases with $D$ as shown in Fig. 3 (b) and Fig. 4 (b).

The scope of Eq. (3) can be extended to include low-energy excitations. One simply creates low-energy excitations in the composite fermion factors $\Phi_{-1}^t(\{z_j^t\})$ and $\Phi_n^b(\{z_j^b\})$ but leaves the flux attachement factors untouched. If one composite fermion is excited from an occupied to an empty orbital in $\Phi_{-1}^t(\{z_j^t\})$ or $\Phi_n^b(\{z_j^b\})$, the resulting wave functions describe neutral excitations since no additional charge is introduced. In Fig. 3 (a) and Fig. 4 (a), these neutral excitations form a dispersive band, which are well captured by the trial wave functions as one can see from the favorable energy values and overlaps. While one naively expects that *both* $\Phi_{-1}^t(\{z_j^t\})$ and $\Phi_n^b(\{z_j^b\})$ contribute to the electronic spectrum, the reality is that *only* the excitations in $\Phi_n^b(\{z_j^b\})$ manifest themselves as neutral excitations of the electrons [see Fig. 2 (b)]. This theoretical prediction can be checked in inelastic light scattering experiments [55, 56]. It is crucial to prove that a system is gapped to ascertain its topological nature. This is done by analyzing the neutral gap defined as the separation between the lowest excited state and the ground state. The evolution of the neutral gap in Fig. 3 (c) and Fig. 4 (c) shows similar trend as the overlaps. Finite-size scaling results in Fig. 3 (d) and Fig. 4 (d) at $D = 0.7$ strongly suggest that the neutral gap saturates to finite values. To have enough data points here, DMRG is employed to compute the gap for several systems that are beyond the reach of ED.

The most surprising feature of the states represented by Eq. (3) is that their charged excitations do not possess fractional charges but obey fractional statistics. This claim implicitly assumes that the total electric charge is conserved, which is important for our discussion but not necessary from the perspective of topological order. To create such entities, we simply add or remove one composite fermion as shown in Fig. 2. For example, if one composite fermion is added to the lowest empty level above the $\Phi_{-1}^t(\{z_j^t\})$ state, the resulting excitation is called a top quasiparticle. The top quasihole, bottom quasiparticle, and bottom quasihole can be defined similarly. The charges carried by them are denoted as $Q_p^t$, $Q_h^t$, $Q_p^b$, and $Q_h^b$. It is helpful to study the consequences of adding or removing one electron when the magnetic fluxes are fixed. This process also changes the effective magnetic fluxes for the composite fermions because flux attachment depends on the number of electrons. If one electron is added to the top monolayer, $\widetilde{N}_\phi^t$ decreases by two units and $\widetilde{N}_\phi^b$ decreases by one unit, so one top quasihole and $n$ bottom quasiparticles are created. If one electron is added to the bottom monolayer, $\widetilde{N}_\phi^t$ decreases by one unit and $\widetilde{N}_\phi^b$ decreases by two units, so one top quasihole and $2n + 1$ bottom quasiparticles are created. It is easy to see that

$$Q_h^t + nQ_p^b = -e, \quad Q_h^t + (2n+1)Q_p^b = -e, \tag{4}$$

which yield $Q_h^t = -e$ and $Q_p^b = 0$ for all $n$. By studying the consequences of removing one electron, we find that $Q_p^t = e$ and $Q_h^b = 0$ for all $n$. Fractional charge is commonly viewed as

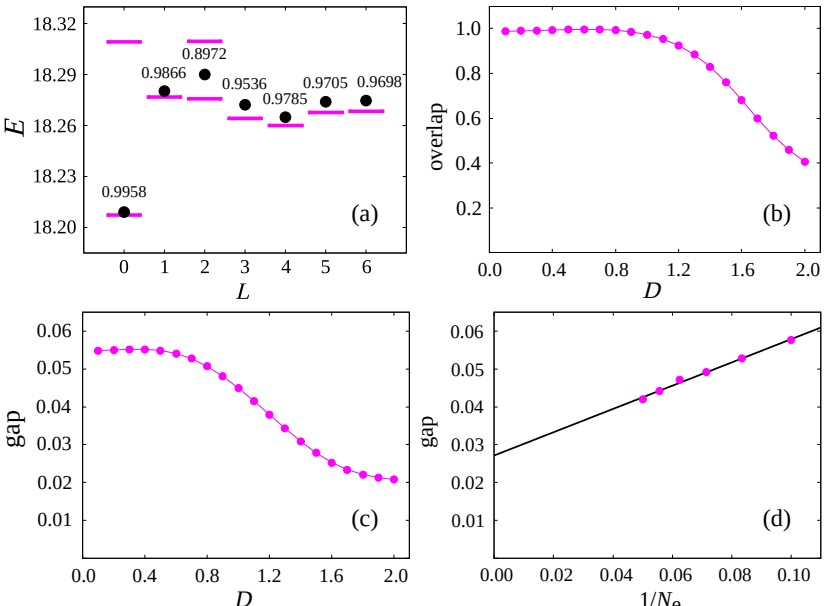

Figure 3: Numerical results about the $\Psi^{221}_{-1,1}$ state. The system parameters are $N^t_e = 6, N^b_e = 6, N^t_\phi = 11, N^b_\phi = 21$ in panels (a-c). (a) The energy spectrum at $D = 0.7$. The exact eigenstates are represented by lines, the trial states are represented by dots, and their overlaps are displayed as numbers. (b) The overlaps between the exact ground state and the trial state at different $D$. (c) The neutral gaps at different $D$. (d) Finite size scaling of the neutral gap versus the total number of electrons $N_e = N^t_e + N^b_e$ at $D = 0.7$.

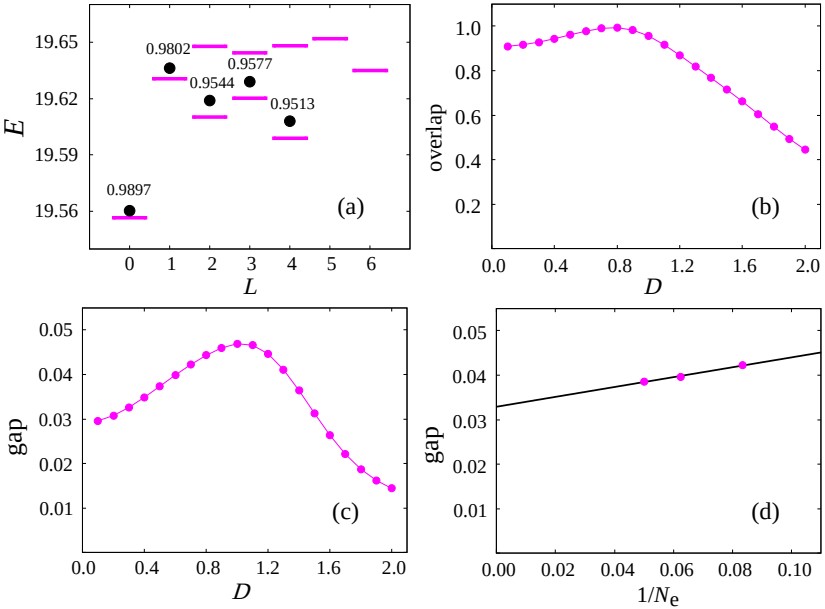

Figure 4: Numerical results about the $\Psi^{221}_{-1,2}$ state. The system parameters are $N^t_e = 6, N^b_e = 6, N^t_\phi = 11, N^b_\phi = 17$ in panels (a-c). (a) The energy spectrum at $D = 0.7$. The symbols are the same as in Fig. 3. (b) The overlaps between the exact ground state and the trial state at different $D$. (c) The neutral gaps at different $D$. (d) Finite size scaling of the neutral gap versus the total number of electrons $N_e = N^t_e + N^b_e$ at $D = 0.7$.

a hallmark of FQH states, and ingenious experimental methods have been designed to verify theoretical predictions [57–60]. It should be possible to demonstrate the *absence* of fractional charge in the $\Psi^{221}_{-1,n}$ states once they are realized.

The information about the charged excitations can be used to derive the Hall conductance of the system using the Laughlin flux insertion argument. For the double monolayer system, two types of measurements have been performed [25, 26]. The first one is the usual Hall conductance $\sigma_{xy}$ when the current passes through both monolayers. The Laughlin argument for this quantity considers the process in which one magnetic flux is adiabatically inserted at the center of both monolayers, which creates one top quasiparticle and $n$ bottom quasihole. The total charge pushed to the boundary of the system is $-e$ so we have $\sigma_{xy} = e^2/h$. Another one is the Hall conductance matrix

$$
\begin{bmatrix}
\sigma^{\text{drive}}_{t,xy} & \sigma^{\text{drag}}_{xy} \\
\sigma^{\text{drag}}_{xy} & \sigma^{\text{drive}}_{b,xy}
\end{bmatrix},
\tag{5}
$$

when the current passes through *only* one monolayer. Its diagonal elements are the drive conductance of the monolayer with current and off-diagonal elements are the drag conductance of the monolayer without current. Their values can be extracted if one considers the process in which one magnetic flux is inserted in *only* one monolayer. To this end, Eqs. 4 should be refined to be

$$
Q^t_h + nQ^b_p = -e_\uparrow, \quad Q^t_h + (2n+1)Q^b_p = -e_\downarrow,
\tag{6}
$$

and similarly for $Q^t_p$ and $Q^b_h$. The subscripts $\uparrow$ and $\downarrow$ are appended to track the orgin of the electron in the sense that the solutions

$$
Q^t_p = \frac{(2n+1)e_\uparrow - ne_\downarrow}{n+1}, \quad Q^b_p = \frac{e_\uparrow - e_\downarrow}{n+1},
$$
$$
Q^t_h = \frac{-(2n+1)e_\uparrow + ne_\downarrow}{n+1}, \quad Q^b_h = \frac{-e_\uparrow + e_\downarrow}{n+1}
\tag{7}
$$

can tell us the "composition" of a charged excitation in terms of the electrons. If one magnetic flux is inserted at the center of the top monolayer, one top quasiparticle is created, which means that $(2n+1)e/(n+1)$ charge is pushed to the boundary in the top monolayer and $-ne/(n+1)$ charge is pushed to the boundary in the bottom monolayer. If one magnetic flux is inserted at the center of the bottom monolayer, $n$ bottom quasihole is created, which means that $-ne/(n+1)$ charge is pushed to the boundary in the top monolayer and $ne/(n+1)$ charge is pushed to the boundary in the bottom monolayer. This analysis yields the Hall conductance matrix

$$
\frac{e^2}{h}
\begin{bmatrix}
\frac{2n+1}{n+1} & -\frac{n}{n+1} \\
-\frac{n}{n+1} & \frac{n}{n+1}
\end{bmatrix},
\tag{8}
$$

and its inverse is the Hall resistance matrix.

The topological properties of Eq. (3) can be described succinctly using the Chern-Simons theory with Lagrangian $\mathcal{L} = \frac{1}{4\pi}\epsilon^{\lambda\mu\nu}K_{IJ}a^I_\lambda\partial_\mu a^J_\nu - a^I_\lambda j^I_\lambda$, where $a^I$ is an emergent gauge field and $j^I$ is the quasiparticle current [61]. The $K$ matrix is the central object in this formalism, which can be motivated in the following manner. If we have two decoupled monolayers with wave functions $\Phi^t_{-1}(\{z^t_j\})$ and $\Phi^b_n(\{z^b_j\})$, its effective theory has a diagonal $K$ matrix with one element being $-1$ and $n$ elements being $1$. The flux attachment is achieved by adding another matrix which has $1$ in its first row and first column (associated with the interlayer flux attachement) and $2$ in other places (associated with the intralayer flux attachment). This yields

$$
K = \begin{pmatrix} 1 & 1 \\ 1 & 3 \end{pmatrix},
\tag{9}
$$

for $n = 1$ and

$$K = \begin{pmatrix} 1 & 1 & 1 \\ 1 & 3 & 2 \\ 1 & 2 & 3 \end{pmatrix}, \tag{10}$$

for $n = 2$. Each excitation of the system is associated with an integer vector $\mathbf{l}$. The charges of the excitations can be probed using a $U(1)$ gauge field $\bar{A}_\mu$ that couples to the excitation current. This results in an additional term $\mathcal{L}_2 = \frac{e}{2\pi\hbar}\epsilon^{\lambda\mu\nu}t_I\bar{A}_\lambda\partial_\mu a_{I\nu}$ in the Lagrangian density with $\mathbf{t}$ called the charge vector. The $U(1)$ charge of an excitation $\mathbf{l}$ is $-e\mathbf{t}^T K^{-1}\mathbf{l}$. The Hall conductance with respect to $\bar{A}_\mu$ is $\sigma_{xy} = e^2\mathbf{t}^T K^{-1}\mathbf{t}/h$. The Chern-Simons theory predicts that the self-statistics angle of the excitation $\mathbf{l}$ is $\theta = \pi\mathbf{l}^T K^{-1}\mathbf{l}$. In contrast to fractional charge, experimental confirmation of fractional braid statistics is much more challenging, but important progresses along this direction have been reported in the past year [62,63]. For the $n = 1$ case, $\mathbf{l}$ is $(\mp 1, 0)^T$ for the top quasiparticle/quasihole and is $(0, \pm 1)^T$ for the bottom quasiparticle/quasihole. For the $n = 2$ case, $\mathbf{l}$ is $(\mp 1, 0, 0)^T$ for the top quasiparticle/quasihole and is $(0, 0, \pm 1)^T$ for the bottom quasiparticle/quasihole. The charges of these excitations and the Hall conductance of the system are reproduced in the field theory formalism. One also concludes that the elementary excitations have fractional braid statistics. The self-statistics angles are $3\pi/2, \pi/2$ for $n = 1$ and $5\pi/3, 2\pi/3$ for $n = 2$.

## 4 Conclusions

In summary, we have studied double monolayer graphene with unequal areas. A class of topological states are proposed and their experimental relevance is investigated. The properties of ground states and elementary excitations are analyzed using numerical calculations, composite fermion theory, and Chern-Simons field theory. The emergence of these states break away from the well-established paradigm that fractional charge and fractional statistics coexist in FQH systems. In addition to the $n = 1, 2$ cases studied above, we believe that the $n = +\infty$ case would also be very interesting. The precise meaning of $n = +\infty$ is that the effective magnetic fluxes for the composite fermions in the bottom monolayer vanish. The simplest state for fermions in zero magnetic field is a Fermi sea, which leads to a non-Fermi liquid of electrons in one-component systems at half filling [35,36]. One can reasonably expect that the same scenario occurs in the two-component $\Psi^{221}_{-1,+\infty}$ state. This paper relies on the existence of electric charge conservation. In general, the interplay between symmetry and topological order is an important topic that is still under investigation [64,65]. The numerical calculations were performed on the sphere, but actual systems have open boundary. It is natural to ask if the difference between the monolayer areas has significant effects. This question calls for a careful analysis of the electron density inhomogeneity and the confinement potential. The answer is likely to depend on the ratio $N_e^t/N_e^b$ and is left for furture studies. We hope that this work would motivate further experiments on double monolayer graphene and related systems.

## Acknowledgements

The author thanks Jize Zhao for sharing his DMRG results. This work was supproted by the NNSF of China under grant No. 11804107.

# A  Hamiltonian Matrix Elements

This Appendix gives the coefficients $F^{\sigma\tau\tau\sigma}_{m_1 m_2 m_4 m_3}$ in the second quantized many-body Hamiltonian. The particles on a sphere experience a radial magnetic field generated by a magnetic monopole at the center. If the magnetic flux through the sphere is $N^\sigma_\phi$, the LLL single-particle wave functions are [42]

$$\psi^{N^\sigma_\phi}_m(\theta, \phi) = \left[ \frac{N^\sigma_\phi + 1}{4\pi} \binom{N^\sigma_\phi}{N^\sigma_\phi - m} \right]^{\frac{1}{2}} u^{N^\sigma_\phi/2 + m} v^{N^\sigma_\phi/2 - m}, \tag{11}$$

where $\theta$ and $\phi$ are the azimuthal and radial angles in the spherical coordinate system, $u = \cos(\theta/2)e^{i\phi/2}, v = \sin(\theta/2)e^{-i\phi/2}$ are spinor coordinates, and $m$ is the $z$ component of the angular momentum. The magnetic length is related to the radius of the sphere by $R^t = \ell_B \sqrt{N^t_\phi/2}$ and $R^b = \ell_B \sqrt{N^b_\phi/2}$. The product of two wave functions can be expanded as

$$\begin{aligned}
\psi^{N^\sigma_\phi}_{m_1} \psi^{N^\tau_\phi}_{m_2} &= (-1)^{N^\sigma_\phi - N^\tau_\phi} \left[ \frac{(N^\sigma_\phi + 1)(N^\tau_\phi + 1)}{4\pi(N^\sigma_\phi + N^\tau_\phi + 1)} \right]^{1/2} \\
&\times \left\langle \frac{N^\sigma_\phi}{2}, -m_1; \frac{N^\tau_\phi}{2}, -m_2 \middle| \frac{N^\sigma_\phi}{2} + \frac{N^\tau_\phi}{2}, -m_1 - m_2 \right\rangle \psi^{N^\sigma_\phi + N^\tau_\phi}_{m_1 + m_2}.
\end{aligned} \tag{12}$$

The Coulomb potential can be expressed using spherical harmonics as

$$\frac{1}{(|\mathbf{r}_1 - \mathbf{r}_2|^2 + D^2)^{1/2}} = \frac{1}{[(R^t)^2 + (R^b)^2 - 2R^t R^b \hat{\mathbf{r}}_1 \cdot \hat{\mathbf{r}}_2 + D^2]^{1/2}} \tag{13}$$

$$= \frac{4\pi}{\sqrt{R^t R^b}} \sum_{L=0}^{+\infty} \sum_{M=-L}^{L} \frac{X^{L+1/2}}{2L+1} \left[\psi^0_{LM}(\theta_1, \phi_1)\right]^* \psi^0_{LM}(\theta_2, \phi_2), \tag{14}$$

where $X$ is the small solution to

$$X^2 - \left[ \frac{(R^t)^2 + (R^b)^2 + D^2}{R^t R^b} \right] X + 1 = 0. \tag{15}$$

These relations help us to obtain

$$F^{\sigma\tau\tau\sigma}_{m_1 m_2 m_4 m_3} = \delta_{m_1 + m_2, m_3 + m_4} \frac{4\pi e^2}{\varepsilon \sqrt{R^t R^b}} \sum_{L=0}^{\min(N^\sigma_\phi, N^\tau_\phi)} \frac{X^{L+1/2}}{2L+1} (-1)^{\frac{N^\sigma_\phi + N^\tau_\phi}{2} - m_1 - m_4} S^1_L S^2_L, \tag{16}$$

where the two coefficients $S^{1,2}_L$ are defined by

$$\left[\psi^{N^\sigma_\phi}_{m_1}\right]^* \left[\psi^0_{LM}\right]^* \psi^{N^\sigma_\phi}_{m_3} = \sum_{L_1=0}^{N^\sigma_\phi} (-1)^{\frac{N^\sigma_\phi}{2} - m_1} S^1_{L_1} \left[\psi^0_{LM}\right]^* \psi^0_{L_1, m_3 - m_1}, \tag{17}$$

$$\left[\psi^{N^\tau_\phi}_{m_2}\right]^* \psi^0_{LM} \psi^{N^\tau_\phi}_{m_4} = \sum_{L_2=0}^{N^\tau_\phi} (-1)^{\frac{N^\tau_\phi}{2} - m_4} S^2_{L_2} \left[\psi^0_{L_2, m_2 - m_4}\right]^* \psi^0_{LM}. \tag{18}$$

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
