# Peer review of "Topological states in double monolayer graphene"

_SciPost Physics, doi:SciPost Phys. 13, 094 (2022)_

## Round 1 · Referee Report · Anonymous (Referee 1) · 2022-3-23

Report

The manuscript by Y.-H. Wu studied a bilayer quantum Hall system motivated by recent experiments on double monolayer graphene quantum Hall effect. The author used various approaches including numerical methods, composite fermion mean field, and Chern-Simons theory to analyze bilayer quantum Hall systems with each layer having different areas. A class of bilayer quantum Hall states was proposed and was found to be a promising candidate ground state of a realistic Hamiltonian. An interesting feature of such quantum Hall states is the existence of anyons that carry integer charge. The manuscript studied an experimentally relevant problem and has interesting discoveries, I am happy to recommend the manuscript after minor revisions are made (see below).

  1. It is not too rare that a topological ordered state has no symmetry fractionalization. Examples I know of are various non-Abelian chiral spin liquids in spin systems with Ising anyons (eg. $SU(2)_2$ in spin-1 system and Kitaev's sixteen fold way), in which the Ising anyon is charge neutral. I think there might be examples of fractional quantum Hall states of fermions as well. It would be good if the author can include some of such references.

  2. On page 3 before Eq.(1), the author assumed that $N_e^t=N_e^b$. Is this assumption realistic for experiments, and how much would the conclusion change if this assumption is relaxed?

  3. I am a bit confused by the analysis of flux attachment in the first paragraph of Sec. 3 Results. The author wrote, "For the top monolayer, $2N_e^t+N_e^b$ fluxes are attached to each electron so $\widetilde N_\phi^t = -N_e^t$, ...". The counting doesn't seem to be right: the total effective flux should be $N_e^t+N_e^b - N_e^t(2N_e^t+N_e^b)$. I guess there is a typo about how many fluxes are attached to each electron?

---

## Round 1 · Referee Report · Anonymous (Referee 2) · 2022-4-7

Report

Inspired by recent experiments on fractional quantum Hall states in double-layer graphene, in this manuscript, the author proposed a new topological state where excitations obey fractional statistics but do not carry a fractional U(1) charge. The model is realized by two graphene layers where electrons cannot hop to the other layer but interact with electrons in the other layer. A key setup to realize the proposed state is that the two layers have different areas such that they can have different fluxes under a uniform magnetic field in the z-direction. I can recommend this work for publication if the authors can address my main concern:
1. Suppose the top layer has a smaller area. Then, in Eq. (1), r1 is restricted in this smaller area (A1) while r2 is not restricted. That means an inhomogeneous Coulomb potential for the bottom layer: r2 inside A1 feels a much stronger Coulomb potential than r2 outside A1. However, in the author's numerical calculation, the two layers are treated as two homocentric spheres with a radius difference of D. Clearly, the bottom layer feels a homogeneous Coulomb potential from the top layer in the numerical setup. My question is, will the inhomogeneity change the true ground state in the realistic system? In the limit where D is much smaller than the sample size, it is hard to believe that all the electrons in the second layer interact equally with the top layer.
2. If a large D comparable to the system size is needed to justify the homogeneity, will it be a reasonable number for the realistic system?
Besides the above concerns, I also have some questions/suggestions:
1. Even though the excitations do not carry a fractional global U(1) charge, they do carry some fractional charge (Eq. (7)) because the system has a U(1)xU(1) symmetry since there are no hoppings between the two layers. Is the fractional statistic stable if one breaks the individual U(1) symmetries?
2. Suppose the U(1)xU(1) symmetry is respected. Is it possible to detect the fractional e_up/e_down from transport measurement inside a single layer?

---

## Round 2 · Referee Report · Anonymous (Referee 2) · 2022-9-2

Report

All my questions have been addressed satisfactorily. I recommend this work for publication in SciPost.

---

## Round 2 · Author Response

Dear Editor,

I would like to thank you for the handling my manuscript.

The two referees made very positive assessment about my work. They also raised some questions and made some comments, which I address below in detail. My reply is somewhat delayed because I have tried to perform additional calculations to answer certain questions.

I hope that you would find that the revised version can be published on SciPost Phys.

Sincerely, Ying-Hai Wu

The manuscript by Y.-H. Wu studied a bilayer quantum Hall system motivated by recent experiments on double monolayer graphene quantum Hall effect. The author used various approaches including numerical methods, composite fermion mean field, and Chern-Simons theory to analyze bilayer quantum Hall systems with each layer having different areas. A class of bilayer quantum Hall states was proposed and was found to be a promising candidate ground state of a realistic Hamiltonian. An interesting feature of such quantum Hall states is the existence of anyons that carry integer charge. The manuscript studied an experimentally relevant problem and has interesting discoveries, I am happy to recommend the manuscript after minor revisions are made (see below).

Reply: I would like to thanks the referee for this very positive overall assessment.

  1. It is not too rare that a topological ordered state has no symmetry fractionalization. Examples I know of are various non-Abelian chiral spin liquids in spin systems with Ising anyons (eg. $SU(2)_2$ in spin-1 system and Kitaev's sixteen fold way), in which the Ising anyon is charge neutral. I think there might be examples of fractional quantum Hall states of fermions as well. It would be good if the author can include some of such references.

    Reply: I thank the referee for this interesting comment. I have not think about the problem in terms of symmetry fractionalization because electric charge conservation is usually assumed when studying FQH states. This is justified in the sense that FQH states was originally observed by measuring electric Hall conductance. Of course, the referee is correct in that this symmetry is not necessary for defining topological order of FQH states. I am not sure if I understand the remark on chiral spin liquids correctly. These states are defined in spin systems, which could have spin-rotation symmetry and/or lattice symmetry. Many papers have discussed symmetry fractionalization in spin liquids, but I am not aware of papers that assign electric charges to spins in this context. In the phrase “charge neutral”, does the author mean charge with respect to some other symmetry? My impression is that people are more interested in the cases with symmetry fractionalization. It is not surprising that sometimes symmetry is not fractionalized but these cases are usually dubbed trivial and do not get special attention. As for FQH states, there are not many works on the properties of symmetry other than electric charge conservation. There are discussions on crystalline symmetry which I now include in the references.

  2. On page 3 before Eq.(1), the author assumed that $N_e^t=N_e^b$. Is this assumption realistic for experiments, and how much would the conclusion change if this assumption is relaxed?

    Reply: The density of electrons in the two monolayers can be controlled in experiments by gates. The cases with balanced and unbalanced monolayers have been studied in previous experiments. Most results reported there are for the balanced cases. This motivates our choice of $N^t_e=N^b_e$. The analysis and calculations can be extended to unbalanced monolayers. Based on previous experience, it is very likely that the states in Eq. (3) can survive to some extent. We have not explored such cases in detail because no essentially new physics is expected, yet the numerical calculations would be very time consuming.

  3. I am a bit confused by the analysis of flux attachment in the first paragraph of Sec. 3 Results. The author wrote, "For the top monolayer, $2N_e^t+N_e^b$ fluxes are attached to each electron so $\widetilde N_\phi^t = -N_e^t$, ...". The counting doesn't seem to be right: the total effective flux should be $N_e^t+N_e^b - N_e^t(2N_e^t+N_e^b)$. I guess there is a typo about how many fluxes are attached to each electron?

    Reply: I thank the referee for pointing out this typo. It should be “For the top monolayer, the total number of fluxes attached to the electrons are $2N^t_e+N^b_e$ so $\widetilde{N}^t_\phi= -N^t_e$, ...”

Inspired by recent experiments on fractional quantum Hall states in double-layer graphene, in this manuscript, the author proposed a new topological state where excitations obey fractional statistics but do not carry a fractional U(1) charge. The model is realized by two graphene layers where electrons cannot hop to the other layer but interact with electrons in the other layer. A key setup to realize the proposed state is that the two layers have different areas such that they can have different fluxes under a uniform magnetic field in the z-direction. I can recommend this work for publication if the authors can address my main concern:

Reply: I would like to thanks the referee for this very positive overall assessment.

  1. Suppose the top layer has a smaller area. Then, in Eq. (1), r1 is restricted in this smaller area (A1) while r2 is not restricted. That means an inhomogeneous Coulomb potential for the bottom layer: r2 inside A1 feels a much stronger Coulomb potential than r2 outside A1. However, in the author's numerical calculation, the two layers are treated as two homocentric spheres with a radius difference of D. Clearly, the bottom layer feels a homogeneous Coulomb potential from the top layer in the numerical setup. My question is, will the inhomogeneity change the true ground state in the realistic system? In the limit where D is much smaller than the sample size, it is hard to believe that all the electrons in the second layer interact equally with the top layer.

  2. If a large D comparable to the system size is needed to justify the homogeneity, will it be a reasonable number for the realistic system?

    Reply: I thank the referee for this very insightful comment. The numerical setup does imply that large D is necessary. In fact, the results suggest that the states of our interest are realized at intermediate D.

    I believe that the essential physics will not be affected by the problem mentioned above for the following reasons.

    Firstly, let me take one step back to consider the Halperin bilayer states. For example, the fermionic 331 state has a parent Hamiltonian under which the state has exactly zero energy. This is valid independent of the geometry (disk, sphere, torus). In most cases, it is assumed that the number of electrons in the two monolayers are the same. However, the state would still have zero energy if the number of electrons in the two monolayers and the areas of the two monolayers are different. This makes me believe that the properties of the states in Eq. (3) would survive to some extent.

    Secondly, one can tune the ratio $N^t_e/N^b_e$ in the two monolayers. This possibility is also mentioned by the other referee. The number of fluxes needed for the states in Eq. (3), which is proportional to the areas of the two monolayers, would change accordingly. While the areas of the two monolayers cannot be exactly same, the inhomogeneity mentioned by the referee can at be significantly mitigated.

    Thirdly, it has been found that edge confinement potential can affect the stability of FQH states. For two monolayers with different areas, a straightforward idea is applying gate voltage in certain parts of the bottom monolayer to stabilize the states in Eq. (3). This actually brings out another question: what is the edge physics? It seems reasonable to conjecture that the top boundary and bottom boundary constitute two edges whose properties are different from previously studied cases.

    With these considerations in mind, I have tried to perform some calculations in the disk geometry which explicitly take into account the effects due to different areas and additional confinement potentials. Unfortunately, this seems to be a quite challenging task and the results are not easy to understand, so it seems much more time is needed. I conclude that a comprehensive discussion should be deferred to future works.

Besides the above concerns, I also have some questions/suggestions:

  1. Even though the excitations do not carry a fractional global U(1) charge, they do carry some fractional charge (Eq. (7)) because the system has a U(1)xU(1) symmetry since there are no hoppings between the two layers. Is the fractional statistic stable if one breaks the individual U(1) symmetries?

    The referee has pointed out an interesting thing. In my discussion, the subscripts in Eq. (7) are only used to keep track of the two monolayers so the Hall conductance matrix can be derived in this way. Each excitation is viewed as a whole object even though it has some components in both monolayers. The fractional statistics does not depend on the existence of two U(1) symmetries. In fact, it is independent of the electric charge conservation based on general principles about topological order. We can again illustrate this point using the Halperin bilayer states, which are usually written in a way with explicit U(1)xU(1) symmetry. It is straightforward to derive analogs of Eq. (7). Adding interlayer tunneling breaks the U(1)xU(1) symmetry, but the states can survive to some extent and the fractional statistics are not destroyed.

  2. Suppose the U(1)xU(1) symmetry is respected. Is it possible to detect the fractional e_up/e_down from transport measurement inside a single layer?

    I am not sure if I understand this comment correctly. Does e_up/e_down means the fractions in Eq. 7? That is, 2n+1/n+1, 1/n+1 etc. These values appear in the Hall conductance matrix, which requires measurement in both layers. I am not aware of any method that can determine such quantities in a single layer.

---

## Round 2 · List of Changes

1. One sentence about the choice $N^t_e=N^b_e$ is added before Eq. (1).

  2. The typo mentioned by one referee is corrected.

  3. In Sec. 4 Conclusions, some discussions on FQH states with symmetry and the inhomogeneity problem is added.

  4. Two references about FQH states with symmetry are added. Two other references about other works on double monolayer graphene are also added.

  5. A few minor changes about the wording are made.

---

## Editorial Decision

published